# Comparison of Three Chimeric Zika Vaccine Prototypes Developed on the Genetic Background of the Clinically Proven Live-Attenuated Japanese Encephalitis Vaccine SA_14_-14-2

**DOI:** 10.3390/ijms26010195

**Published:** 2024-12-29

**Authors:** Byung-Hak Song, Jordan C. Frank, Sang-Im Yun, Justin G. Julander, Jeffrey B. Mason, Irina A. Polejaeva, Christopher J. Davies, Kenneth L. White, Xin Dai, Young-Min Lee

**Affiliations:** 1Department of Animal Dairy and Veterinary Sciences, College of Agriculture and Applied Sciences, Utah State University, Logan, UT 84322, USA; byunghaksong1004@gmail.com (B.-H.S.); jcfrank.v@gmail.com (J.C.F.); sangim.yun@usu.edu (S.-I.Y.); justin.julander@usu.edu (J.G.J.); irina.polejaeva@usu.edu (I.A.P.); chris.davies@usu.edu (C.J.D.); ken.white@usu.edu (K.L.W.); 2Institute for Antiviral Research, Utah State University, Logan, UT 84322, USA; 3Department of Veterinary Clinical and Life Sciences, College of Veterinary Medicine, Center for Integrated BioSystems, Utah State University, Logan, UT 84322, USA; jeff.mason@usu.edu; 4Utah Agricultural Experiment Station, Utah State University, Logan, UT 84322, USA; xin.dai@usu.edu

**Keywords:** Zika virus, Japanese encephalitis virus, orthoflavivirus, SA_14_-14-2, live-attenuated vaccine, chimeric vaccine

## Abstract

Zika virus (ZIKV) is a medically important mosquito-borne orthoflavivirus, but no vaccines are currently available to prevent ZIKV-associated disease. In this study, we compared three recombinant chimeric viruses developed as candidate vaccine prototypes (rJEV/ZIKV^MR-766^, rJEV/ZIKV^P6-740^, and rJEV/ZIKV^PRVABC-59^), in which the two neutralizing antibody-inducing prM and E genes from each of three genetically distinct ZIKV strains were used to replace the corresponding genes of the clinically proven live-attenuated Japanese encephalitis virus vaccine SA_14_-14-2 (rJEV). In WHO-certified Vero cells (a cell line suitable for vaccine production), rJEV/ZIKV^P6-740^ exhibited the slowest viral growth, formed the smallest plaques, and displayed a unique protein expression profile with the highest ratio of prM to cleaved M when compared to the other two chimeric viruses, rJEV/ZIKV^MR-766^ and rJEV/ZIKV^PRVABC-59^, as well as their vector, rJEV. In IFNAR^−/−^ mice, an animal model of ZIKV infection, subcutaneous inoculation of rJEV/ZIKV^P6-740^ caused a low-level localized infection limited to the spleen, with no clinical signs of infection, weight loss, or mortality; in contrast, the other two chimeric viruses and their vector caused high-level systemic infections involving multiple organs, consistently leading to clear clinical signs of infection, rapid weight loss, and 100% mortality. Subsequently, subcutaneous immunization with rJEV/ZIKV^P6-740^ proved highly effective, offering complete protection against a lethal intramuscular ZIKV challenge 28 days after a single-dose immunization. This protection was specific to ZIKV prM/E and likely mediated by neutralizing antibodies targeting ZIKV prM/E. Therefore, our data indicate that the chimeric virus rJEV/ZIKV^P6-740^ is a highly promising vaccine prototype for developing a safe and effective vaccine for inducing neutralizing antibody-mediated protective immunity against ZIKV.

## 1. Introduction

Zika virus (ZIKV) is a clinically important member of the genus *Orthoflavivirus* in the family *Flaviviridae* [1,2]. Phylogenetically, ZIKV is closely related to other globally significant pathogenic orthoflaviviruses, such as Japanese encephalitis virus (JEV), West Nile virus (WNV), dengue virus (DENV), and yellow fever virus (YFV) [3]. Historically, ZIKV was first isolated from a rhesus macaque monkey in Uganda in 1947, and this African strain was named MR-766 [4]. For decades after its discovery, ZIKV was endemic primarily in the tropical countries of Africa and Asia [5]; the first Asian strain, designated P6-740, was isolated from a collection of *Aedes aeqypti* mosquitoes in Malaysia in 1966 [6]. Over the past two decades, however, not only have sporadic cases of ZIKV infection been reported in the previously recognized endemic countries in Africa and Asia, but several large outbreaks have also occurred outside these regions [7]: (i) the first on Yap Island in the northwestern Pacific Ocean in 2007 [8,9,10]; (ii) the next in French Polynesia in the southeastern Pacific Ocean in 2013–2014 [11,12], spreading to other southern Pacific Islands [13,14,15]; and (iii) the most recent in Brazil in South America in 2015–2016 [16,17,18], spreading rapidly throughout the Americas [19,20,21]. At the beginning of the 2015–2016 outbreak, the American epidemic strain, designated PRVABC-59, was isolated from a human patient in Puerto Rico and appeared to have originated from an ancestor of the Asian lineage [22,23]. Since the 2015–2016 epidemic, ZIKV has continued to circulate in a number of tropical and subtropical countries throughout the globe at low levels [24,25]. Thus, the continuous evolution of ZIKV and the persistent expansion of its activity pose a challenge to global public health [26,27,28].

ZIKV is a mosquito-borne arbovirus with two natural transmission cycles: a sylvatic cycle occurring between non-human primates via arboreal mosquitoes and an urban cycle occurring between humans via urban mosquitoes [7]. Incidentally, ZIKV is also believed to be transmitted from non-human primates to humans via arboreal mosquitoes [7]. In ZIKV-affected areas, the virus has been isolated from a large number of mosquito species mostly belonging to the genus *Aedes*, which are potentially capable of acting as vectors for viral transmission [26,29]. Among the *Aedes* mosquitoes, the transmission of ZIKV, particularly in an urban cycle, is believed to be largely mediated by two specific species [30]: *A. aegypti*, serving as the primary vector [31,32,33,34], and *A. albopictus*, serving as the secondary vector [35,36,37]. In addition to the main mosquito-borne transmission, ZIKV can be transmitted without a mosquito vector from person to person through sexual contact [38,39,40] or blood transfusion [41,42], from a pregnant woman to her fetus during pregnancy [43,44,45,46,47], and possibly from a mother to her child after delivery through breastfeeding [48,49,50,51,52].

In 1954, ZIKV was first documented to cause jaundice in humans in Nigeria [53]. Between 1954 and the 2007 outbreak, only a handful of benign cases were reported in endemic countries of Africa and Asia [5,19]. During the 2007 outbreak, ZIKV caused a mild illness marked by fever, arthralgia, rash, and conjunctivitis [8,9]. Since then, however, ZIKV has been linked to severe neurological syndromes [54,55,56]: (i) Guillain–Barré syndrome in adults (first observed during the 2013–2014 outbreak [57,58] and later during the 2015–2016 outbreak [59,60]), which results in immune-mediated impairment of the peripheral nerves [61,62]; and (ii) congenital Zika syndrome in newborns (first noted during the 2015–2016 outbreak [63,64,65,66,67]), which results in microcephaly, brain abnormalities, and other developmental issues [68,69,70]. Retrospective studies conducted after the 2013–2014 outbreak have also shown an increase in cases of microcephaly and other fetal abnormalities [71,72]. Despite these clinically significant ZIKV-associated disease outcomes and lifelong neurological sequelae, there are no available vaccines or drugs for its prevention or treatment [73,74,75,76].

Like all orthoflaviviruses, ZIKV has a linear, 11 kb, single-stranded, positive-sense RNA genome, which is capped at the 5′ end but non-polyadenylated at the 3′ end [3,23]. The genome contains one long open reading frame (ORF) situated between two short 5′ and 3′ non-coding regions (NCRs), each of which includes an array of cis-acting RNA elements important for regulating viral translation, RNA replication, and potentially other steps of viral replication [77,78,79,80,81]. The single ORF is translated into a polyprotein, which is cleaved both co-translationally and post-translationally into three structural proteins (capsid [C], pre-membrane [prM], and envelope [E]) and seven nonstructural proteins (NS1, NS2A, NS2B, NS3, NS4A, NS4B, and NS5) [82,83]. The latter seven nonstructural proteins primarily provide enzymatic activity or non-enzymatic components of viral replication, particularly related to polyprotein processing, RNA replication, and particle morphogenesis [84,85,86,87,88]. These seven proteins also function as antagonists in host cell responses to viral replication [89,90,91,92]. The three structural proteins are part of an infectious virion [93,94,95]. The infectious ZIKV is an enveloped particle approximately 50 nm in diameter. It possesses a nucleocapsid core, which is a ribonucleocomplex of the genomic RNA and 60 C:C homodimers [96,97]. This nucleocapsid is enclosed in a lipid bilayer with 90 (M:E)_2_ heterodimers embedded in the viral envelope membrane, with the E proteins forming a smooth outermost protein shell and the M proteins buried underneath it [98,99]. It is noteworthy that prM, the precursor of M, acts as a chaperone to assist in the proper folding and oligomerization of E [100], which in turn directs viral entry into a host cell [101,102] and induces the production of neutralizing antibodies in the host organism [103,104]. For vaccine development, a large body of data has indicated that neutralizing antibodies against the two surface proteins prM and E provide protective immunity and serve as an important correlate for protection against ZIKV [105,106,107,108,109,110,111,112,113,114,115,116]. Thus, the prM and E proteins are the primary targets that induce protective neutralizing antibody responses during ZIKV infection.

In the present study, we used the clinically proven live-attenuated JEV vaccine SA_14_-14-2 [117,118], licensed for human use [119,120], as a vaccine vector to create three recombinant chimeric viruses as candidate vaccine prototypes against ZIKV, in which the prM and E genes of JEV SA_14_-14-2 were replaced with the corresponding genes from each of three genetically distinct ZIKV strains. We then compared the virological properties of the three chimeric viruses in cell culture and assessed their safety, immunogenicity, and protective efficacy in mice to identify the most desirable live-attenuated chimeric vaccine prototype for preventing ZIKV infection. Among the three chimeric viruses, we identified the one that is the most promising vaccine prototype for ZIKV, maximizing vaccine safety, ensuring protective efficacy, and capitalizing on the advantage of single-dose immunization.

## 2. Results

### 2.1. Rationale for Selecting the Live-Attenuated JEV SA_14_-14-2 as a Vaccine Platform and Three Genetically Distinct Strains of ZIKV

To develop a promising live chimeric vaccine prototype for ZIKV, we chose the live-attenuated JEV vaccine SA_14_-14-2 as the genetic backbone because it is widely administered to children in many JEV-endemic countries [120] and has an excellent safety and efficacy profile [119]. Also, JEV is phylogenetically closely related to ZIKV within the *Orthoflavivirus* genus [1,2,3]. For genetic manipulation of this vaccine virus, we previously developed its full-length infectious cDNA clone [121], allowing us to create recombinant chimeric viruses expressing the two neutralizing antibody-inducible, functional prM and E proteins of ZIKV. We hypothesized that the genetic sequence of the prM and E genes of ZIKV would determine the degree of attenuation of the chimeric viruses since these gene products form the viral surface protein complex [98,99], a key determinant of orthoflavivirus virulence [100,101]. To test our hypothesis, we aimed to generate three chimeric viruses by replacing the prM and E genes of JEV SA_14_-14-2 with the corresponding genes from each of three spatiotemporally distinct and genetically divergent ZIKV strains [23,82]: African MR-766 (Uganda, 1947), Asian P6-740 (Malaysia, 1966), and American PRVABC-59 (Puerto Rico, 2015). Using these three ZIKV strains in the present study was advantageous because their RNA genomes are fully sequenced [23], their full-length infectious cDNA clones have been successfully constructed [82], and their cDNA-derived recombinant ZIKVs are well characterized in terms of their virological characteristics in cell culture and pathogenic properties in mice [82].

### 2.2. Generation of Three Chimeric JEV/ZIKVs Expressing the Functional prM and E Proteins of Three Genetically Distinct ZIKV Strains

To produce three chimeric JEV/ZIKVs, we utilized our reverse genetics system based on the full-length infectious cDNA clone of JEV SA_14_-14-2, designated pBac/JEV (Figure 1A). First, we constructed three full-length chimeric cDNA constructs, replacing the prM and E genes of JEV SA_14_-14-2 with the corresponding genes of ZIKV MR-766, P6-740, and PRVABC-59, designated pBac/JEV/ZIKV^MR-766^, pBac/JEV/ZIKV^P6-740^, and pBac/JEV/ZIKV^PRVABC-59^, respectively (Figure 1A). As sources for the prM-E gene segment of ZIKV, we used our three full-length infectious ZIKV cDNA clones, designated pBac/ZIKV MR-766, pBac/ZIKV P6-740, and pBac/ZIKV PRVABC-59 (Figure 1A). The 5′ and 3′ ends of the prM-E gene segment were predicted using the SignalP program [122] to identify the location of signal peptide cleavage sites at the C-prM and E-NS1 junctions of the viral polyprotein in the three ZIKV strains. This was done in conjunction with the Clustal W program [123] to compare the amino acid sequences at those junctions in the viral polyprotein of the three ZIKV strains with those of JEV SA_14_-14-2.

Next, we examined the functionality of synthetic RNAs transcribed in vitro from the three full-length chimeric JEV/ZIKV cDNAs, comparing them to the parental JEV cDNA and to three original ZIKV cDNAs by measuring their specific infectivity in plaque-forming units (PFUs) per μg after RNA transfection into WHO Vero RCB 10–87 cells (hereafter referred to as Vero cells [124]). We chose this Vero cell line because it is suitable for human vaccine production and susceptible to both JEV and ZIKV. We found that the RNAs transcribed from each of the three chimeric JEV/ZIKV cDNAs were as functional as those transcribed from the parental JEV cDNA and each of the three original ZIKV cDNAs, with an estimated infectivity of 5.1–8.8 × 10^5^ PFUs/μg (Figure 1B). In the case of all seven cDNAs, we recovered the molecularly cloned cDNA-derived recombinant viruses from RNA-transfected cells at 72 h post-transfection (hpt), with virus titers ranging from 5.5 × 10^4^ to 2.0 × 10^7^ PFUs/mL (Figure 1C). Intriguingly, we noted that among the three chimeric viruses, rJEV/ZIKV^P6-740^ (derived from pBac/JEV/ZIKV^P6-740^) had the slowest rate of virus accumulation, starting from 6.0 × 10^2^ PFUs/mL at 24 hpt and reaching 5.5 × 10^4^ PFUs/mL at 72 hpt; in contrast, rJEV/ZIKV^MR-766^ (derived from pBac/JEV/ZIKV^MR-766^) and rJEV/ZIKV^PRVABC-59^ (derived from pBac/JEV/ZIKV^PRVABC-59^) started from 4.0–5.0 × 10^3^ PFUs/mL at 24 hpt and reached 1.7–1.8 × 10^6^ PFUs/mL at 72 hpt. In addition, we found that rJEV/ZIKV^P6-740^ behaved more like the parental rJEV (derived from pBac/JEV), although the parental rJEV accumulated slightly faster, starting from 3.0 × 10^3^ PFUs/mL at 24 hpt and reaching 1.0 × 10^5^ PFUs/mL at 72 hpt. This difference in virus accumulation rate was also observed among the three original ZIKVs. Specifically, rZIKV P6-740 (derived from pBac/ZIKV P6-740) had a slower rate of virus accumulation than either rZIKV MR-766 (derived from pBac/ZIKV MR-766) or rZIKV PRVABC-59 (derived from pBac/ZIKV PRVABC-59); thus, our results indicate that the prM and E genes of ZIKV P6-740 are responsible for the low virus production in both the chimeric JEV/ZIKV and original ZIKV contexts.

### 2.3. Comparison of the Virological Properties of the Three Chimeric JEV/ZIKVs in Vero Cells

To determine whether genetic variations in the prM and E genes of ZIKV affect the replication capabilities of the three chimeric JEV/ZIKVs (rJEV/ZIKV^MR-766^, rJEV/ZIKV^P6-740^, and rJEV/ZIKV^PRVABC-59^) in cell culture, we infected Vero cells with each chimeric virus or the parental JEV (rJEV) for comparison at a multiplicity of infection (MOI) of 1. We then examined their virological properties, including the expression of the prM and E proteins, viral growth kinetics, and plaque morphology (Figure 2). For reference, we also included the three original ZIKVs (rZIKV MR-766, rZIKV P6-740, and rZIKV PRVABC-59).

First, we examined the expression of viral prM and E proteins by immunoblotting total cell lysates collected at 24 h post-infection (hpi) with a panel of JEV- or ZIKV-specific polyclonal rabbit antisera, each recognizing the viral proteins C, prM/M, E, or NS1 (Figure 2A). The results showed that all three chimeric JEV/ZIKVs expressed the replaced ZIKV prM and E genes as expected. However, we made two interesting observations: (i) The ratio of the uncleaved prM to its cleaved M protein differed dramatically from one chimeric JEV/ZIKV to another. The highest ratio was detected in the rJEV/ZIKV^P6-740^-infected cells, an intermediate ratio in the rJEV/ZIKV^PRVABC-59^-infected cells, and the lowest ratio in the rJEV/ZIKV^MR-766^-infected cells. This pattern was also mirrored in the cells infected with each of the three original ZIKVs, showing the highest ratio in the rZIKV P6-740-infected cells, an intermediate ratio in the rZIKV PRVABC-59-infected cells, and the lowest ratio in the rZIKV MR-766-infected cells. For reference, we also checked the prM-to-M ratio in rJEV-infected cells and saw the lowest ratio in these cells, resembling that seen in rJEV/ZIKV^MR-766^-infected cells. (ii) We found that the mobility of the ZIKV E protein expressed in the rJEV/ZIKV^P6-740^-infected cells on a reducing gel was slightly faster than the same protein expressed in both the rJEV/ZIKV^MR-766^-infected and rJEV/ZIKV^PRVABC-59^-infected cells. Among the three original ZIKVs, this faster mobility of the ZIKV E protein was also detected in cells infected with rZIKV P6-740, but not in those infected with rZIKV MR-766 or rZIKV PRVABC-59.

Second, we used immunoplaque assays on Vero cells to evaluate the kinetics of viral growth over 3 days after infection by titrating culture supernatants collected at 6, 12, 18, 24, 36, 48, 60, and 72 hpi (Figure 2B). We found that among the three chimeric viruses, rJEV/ZIKV^P6-740^ showed the slowest growth kinetics, reaching a maximum titer of 9.0 × 10^5^ PFUs/mL at 72 hpi. In contrast, both rJEV/ZIKV^MR-766^ and rJEV/ZIKV^PRVABC-59^ exhibited similar growth kinetics, with virus titers ~1 log higher than rJEV/ZIKV^P6-740^ at all time points after 24 hpi, reaching maximum titers of 5.9 × 10^6^ and 6.5 × 10^6^ PFUs/mL at 72 hpi, respectively. Notably, rJEV/ZIKV^P6-740^ grew less efficiently than the parental rJEV, which had virus titers ~0.5 log higher than rJEV/ZIKV^P6-740^ at all time points after 24 hpi and reached a maximum titer of 2.5 × 10^6^ PFUs/mL at 72 hpi. Among the three original ZIKVs, rZIKV P6-740 showed the slowest growth kinetics, reaching a maximum titer of 6.0 × 10^6^ PFUs/mL at 72 hpi, as compared to rZIKV MR-766 and rZIKV PRVABC-59, which reached maximum titers of 2.0 × 10^7^ and 6.0 × 10^6^ PFUs/mL at 48 hpi, respectively, one day earlier than rZIKV P6-740.

Third, we analyzed the morphology of viral plaques by immunostaining cell monolayers at 5 days post-infection with a polyclonal rabbit α-JEV NS3 antiserum in the case of the parental JEV and three chimeric JEV/ZIKVs, or a polyclonal rabbit α-ZIKV NS1 antiserum in the case of the three original ZIKVs (Figure 2C). The results showed that the three chimeric JEV/ZIKVs formed a homogeneous population of various plaque sizes. Specifically, rJEV/ZIKV^P6-740^ displayed the smallest plaques (2.0 mm in diameter), rJEV/ZIKV^PRVABC-59^ displayed medium-sized plaques (4.9 mm), and rJEV/ZIKV^MR-766^ displayed the largest plaques (5.6 mm). Similar differences in plaque sizes were observed with the three original ZIKVs: rZIKV P6-740 (4.0 mm), rZIKV PRVABC-59 (5.6 mm), and rZIKV MR-766 (6.0 mm). In comparison, the parental rJEV formed medium-sized plaques (4.2 mm), which were larger than those formed by rJEV/ZIKV^P6-740^ (2.0 mm) but smaller than those formed by rJEV/ZIKV^PRVABC-59^ (4.9 mm) and rJEV/ZIKV^MR-766^ (5.6 mm).

Collectively, our data demonstrate that rJEV/ZIKV^P6-740^ has several unique virological characteristics in Vero cells as compared to rJEV/ZIKV^MR-766^ and rJEV/ZIKV^PRVABC-59^. Specifically, rJEV/ZIKV^P6-740^ showed the highest prM-to-M protein expression ratio, displayed the slowest growth kinetics, and formed the smallest plaques. These characteristics are associated with the replaced prM and E genes of rZIKV P6-740.

### 2.4. Comparison of the Attenuation Phenotypes of the Three Chimeric JEV/ZIKVs in IFNAR^−/−^ Mice

To determine whether genetic variations in the prM and E genes of ZIKV affect the attenuation phenotypes of the three chimeric JEV/ZIKVs (rJEV/ZIKV^MR-766^, rJEV/ZIKV^P6-740^, and rJEV/ZIKV^PRVABC-59^) in mice, we initially tested the vaccine candidates in wild-type weanling C57BL/6J mice (Figure 3). Groups of 4-week-old mice (*n* = 6 per group; three of each sex) were infected subcutaneously with a maximum dose of 1.5 × 10^5^ PFUs/mouse of each chimeric virus. In parallel, we included the parental JEV (rJEV) for comparison and the three original ZIKVs (rZIKV MR-766, rZIKV P6-740, and rZIKV PRVABC-59) as references. As a negative control, six mice (three of each sex) were mock-infected subcutaneously with culture supernatant collected from uninfected Vero cells. For 22 days post-infection, the mice were monitored daily for clinical signs (e.g., decreased activity, ruffled fur, hunched posture, tremors, and hind limb paralysis), body weight changes, and survival rates (Figure 3A). None of the mice infected with the three chimeric JEV/ZIKVs died, nor did any of those receiving the parental JEV or any of the three original ZIKVs (Figure 3B). However, in the case of all seven viruses, although clear clinical signs were not observed, we noted subtle differences in body weight gains over time (Figure 3C). Specifically, of the three chimeric JEV/ZIKVs, rJEV/ZIKV^P6-740^-infected mice showed an average weight gain over time that was essentially identical to that of the mock-infected mice (*p* = 0.45) and to that of the rJEV-infected mice (*p* = 0.68). In contrast, both rJEV/ZIKV^MR-766^-infected and rJEV/ZIKV^PRVABC-59^-infected mice showed an average weight gain ~10% lower than that of the rJEV/ZIKV^P6-740^-infected mice (*p* = 0.07). All three original ZIKVs were associated with a minor decline in the average weight gain over time, with rZIKV MR-766 causing the most noticeable reduction (18% lower than the mock-infected control, *p* = 0.03), rZIKV P6-740 causing an intermediate reduction (15% lower than the mock-infected control, *p* = 0.13), and rZIKV PRVABC-59 causing the least reduction (9% lower than the mock-infected control, *p* = 0.33).

Next, we analyzed the attenuation phenotypes of the three prM-E gene-replaced chimeric JEV/ZIKVs by testing them in age-matched interferon-α/β receptor-knockout (IFNAR^−/−^) mice on the C57BL/6J genetic background [82] (Figure 4A), as we had done earlier in wild-type weanling C57BL/6J mice. Interestingly, of the three chimeric JEV/ZIKVs, rJEV/ZIKV^P6-740^ was the only one that produced no clinical signs or mortality in IFNAR^−/−^ mice when infected subcutaneously at a maximum dose of 1.5 × 10^5^ PFUs/mouse (Figure 4B). Also, all six rJEV/ZIKV^P6-740^-infected mice gained body weight over the entire 22-day period after infection, as did the mock-infected mice (Figure 4C). In contrast, the other two chimeric JEV/ZIKVs (rJEV/ZIKV^MR-766^ and rJEV/ZIKV^PRVABC-59^) produced clear clinical signs (e.g., decreased activity, ruffled fur, hunched posture, tremors, and/or hind limb paralysis), resulting in 100% mortality with estimated survival times of 6.2 and 6.9 days, respectively, similar to the parental JEV (rJEV), which also had 100% mortality with an estimated survival time of 5.8 days (Figure 4B). Among the three original ZIKVs, both rZIKV MR-766 and rZIKV P6-740 produced typical clinical signs (e.g., decreased activity, ruffled fur, hunched posture, tremors, and/or hind limb paralysis) and caused 100% mortality, with estimated survival times of 6 days for rZIKV MR-766 and 9 days for rZIKV P6-740 (Figure 4B). In contrast, the rZIKV PRVABC-59 virus was associated with 50% mortality, with an estimated survival time of 10 days in the mice that died (Figure 4B). In the mice that survived, this virus produced only mild clinical signs (e.g., decreased activity and/or ruffled fur) but caused a significant reduction in body weight (Figure 4C). Specifically, on day 10, when the surviving rZIKV PRVABC-59-infected mice started to recover, their body weight was estimated to be 94% of the initial weight, compared to 114% for both the mock-infected and rJEV/ZIKV^P6-740^-infected mice. The body weight differences were even larger at the end of the experiment on day 22: 95% of the initial weight for the recovered rZIKV PRVABC-59-infected mice, and 123% and 131% for the mock-infected and rJEV/ZIKV^P6-740^-infected mice, respectively. The weight reductions caused by rZIKV PRVABC-59 were significant, with *p*-values of 0.01 on day 10 and 0.002 on day 22.

In IFNAR^−/−^ mice, we further evaluated viral loads in various organs (brain, spleen, lung, kidney, ovary/testis, heart, and pancreas) at 2, 4, and 6 days after subcutaneous inoculation of 12 mice (6 of each sex) at 4 weeks of age with 1.5 × 10^5^ PFUs/mouse of each of the three chimeric JEV/ZIKVs, parental JEV for comparison, and three original ZIKVs as references (Figure 4A). At each time point, four mice (two of each sex) were sacrificed to collect the organs for virus titration. For all seven viruses, except for rJEV/ZIKV^P6-740^, various levels of viral replication ranging from ~10 to ~10^9^ PFUs/organ were detected in all the tested organs (Figure 4D). In the case of rJEV/ZIKV^P6-740^, no detectable level of productive viral replication was observed in any organ except the spleen, which showed a relatively low level of viral replication, producing less than ~10^4^ PFUs/organ (Figure 4D).

Based on our infection experiments in wild-type weanling C57BL/6J and IFNAR^−/−^ mice, we concluded that of the three chimeric JEV/ZIKVs, rJEV/ZIKV^P6-740^ was more attenuated than rJEV/ZIKV^MR-766^ or rJEV/ZIKV^PRVABC-59^. Therefore, we selected rJEV/ZIKV^P6-740^ as a candidate vaccine prototype against ZIKV infection for our next immunogenicity and protective efficacy experiments.

### 2.5. Evaluation of the Immunogenicity and Protective Efficacy of the Chimeric Virus rJEV/ZIKV^P6-740^ as a Candidate Vaccine Prototype Against ZIKV Infection

To test whether rJEV/ZIKV^P6-740^ can induce neutralizing antibodies and provide protection against ZIKV infection, we conducted two experiments: one to assess the neutralizing antibody response and the other to evaluate protective efficacy in weanling IFNAR^−/−^ mice (Figure 5A). For the assessment of the neutralizing antibody response, groups of 4-week-old mice (*n* = 6 per group; half male and half female) were either mock-immunized or immunized subcutaneously with 1.5 × 10^5^ PFUs/mouse of rJEV/ZIKV^P6-740^. Blood samples were collected before immunization and 28 days after to determine whether the immunization induced neutralizing antibodies against each of the three ZIKV strains (rZIKV MR-766, rZIKV P6-740, and rZIKV PRVABC-59) or against JEV SA_14_, a parental wild-type virulent strain of JEV SA_14_-14-2 (for comparison). For the evaluation of protective efficacy, the initial immunization step was conducted as described above. On day 28 post-immunization, groups of mock-immunized or rJEV/ZIKV^P6-740^-immunized mice (*n* = 6 per group; half male and half female) were mock-challenged or challenged intramuscularly with 1.5 × 10^5^ PFUs/mouse of each of the three ZIKV strains or JEV SA_14_. Following the challenge, the mice were monitored daily for clinical signs, body weight changes, and survival rates for 22 days to determine the protective efficacy of the immunization.

First, we assessed the level of neutralizing antibodies induced by rJEV/ZIKV^P6-740^ immunization using a 50% plaque reduction neutralization test (PRNT_50_) on Vero cells with pre-immune and post-immune sera (Figure 5B). Our data showed that the post-immune sera neutralized both the homologous prM-E expressing rZIKV P6-740 and the heterologous prM-E expressing rZIKV MR-766 and rZIKV PRVABC-59 equally well, with log_10_ PRNT_50_ values ranging from 1.81 to 3.01 and median values between 2.56 and 2.71. As expected, the pre-immune sera did not neutralize any of the three original ZIKVs. However, unlike the three original ZIKVs, the post-immune sera failed to neutralize JEV SA_14_. Thus, our results demonstrate that a single-dose immunization of rJEV/ZIKV^P6-740^ in IFNAR^−/−^ mice induces a high titer of neutralizing antibodies reactive with all three genetically distinct ZIKV strains, but not with the phylogenetically related JEV SA_14_.

Second, we evaluated the protective efficacy of rJEV/ZIKV^P6-740^ immunization against challenges with each of the three ZIKV strains or JEV SA_14_. In the three mock-immunized ZIKV-challenged control groups, our data revealed differences in mouse susceptibility to infection by the three ZIKV strains (Figure 5C,D). Specifically, all the mock-immunized rZIKV MR-766-challenged mice showed clear clinical signs (e.g., decreased activity, ruffled fur, hunched posture, tremors, and/or hind limb paralysis), resulting in rapid weight loss and 100% mortality within 6–7 days post-challenge. In contrast, all the mock-immunized rZIKV P6-740- or PRVABC-59-challenged mice remained alive but exhibited mild clinical signs (e.g., decreased activity and/or ruffled fur), leading to transient weight loss, reaching maximum loss at day 7 (for rZIKV PRVABC-59) or day 8 (for rZIKV P6-740) post-challenge, with partial recovery afterward, compared to all the mock-immunized mock-challenged control mice. In more detail, on day 7, the recovering mock-immunized rZIKV P6-740- and PRVABC-59-challenged mice weighed 84% and 87% of their initial weights, significantly lower than that of the mock-immunized mock-challenged control mice, which had gained an estimated 2% more than their initial weights (*p* < 0.0001). The significance of weight reduction continued through the remaining days of the experiment. When the experiment ended on day 22, the body weight of the mock-immunized mock-challenged control mice was estimated at 111% of their initial weight, but the recovered mock-immunized rZIKV P6-740- and PRVABC-59-challenged mice had just regained 97% and 100% of their initial weights, respectively.

In the three rJEV/ZIKV^P6-740^-immunized, ZIKV-challenged experimental groups, our results demonstrated complete protection against challenges with each of the three ZIKV strains (Figure 5C,D). Specifically, all the rJEV/ZIKV^P6-740^-immunized rZIKV MR-766-challenged mice were completely protected, showing no clinical signs, no weight loss, and no mortality, in stark contrast to the mock-immunized rZIKV MR-766-challenged control mice. Similarly, all the rJEV/ZIKV^P6-740^-immunized rZIKV P6-740- or PRVABC-59-challenged mice were also fully protected, showing no clinical signs and no weight loss, in contrast to the mock-immunized rZIKV P6-740- or PRVABC-59-challenged control mice. Overall, the three rJEV/ZIKV^P6-740^-immunized, ZIKV-challenged experimental groups all showed nearly identical weight gains of about 10% by the end of the experiment on day 22 (*p* > 0.40). However, when challenged with rJEV SA_14_, both the mock-immunized and the rJEV/ZIKV^P6-740^-immunized mice developed equally clear clinical signs, resulting in rapid weight loss and 100% mortality within 5–8 days post-challenge. Thus, our findings demonstrate that a single-dose immunization with rJEV/ZIKV^P6-740^ in IFNAR^−/−^ mice provides ZIKV prM/E-specific full protection against both lethal and non-lethal challenges with all three genetically distinct ZIKV strains.

### 2.6. Comparison of the Amino Acid Sequence of the prM and E Proteins of Three Genetically Distinct ZIKV Strains

To gain insight into the unique biological properties of rJEV/ZIKV^P6-740^, relative to rJEV/ZIKV^MR-766^ and rJEV/ZIKV^PRVABC-59^, we performed a multiple sequence alignment for phylogenetic analysis and sequence comparison using the amino acid sequences of the prM and E proteins of the three ZIKV strains (MR-766, P6-740, and PRVABC-59). First, our phylogenetic analysis showed that the P6-740 strain is closely related to the PRVABC-59 strain but distantly related to the MR-766 strain (Figure 6A). This result, based on the amino acid sequence of the prM and E proteins, aligns with our previous report describing the genetic relationship between the three ZIKV strains based on their complete genome sequences [82]. It indicates that the MR-766 strain belongs to the African lineage, whereas both the P6-740 and PRVABC-59 strains belong to the Asian lineage, with PRVABC-59 descending from a predecessor in the Asian lineage. Second, our pairwise amino acid sequence comparison revealed that the prM and E proteins of P6-740 have a sequence divergence of 3.2%, differing by 21 amino acid residues (9 in prM and 12 in E) from those of MR-766, and a sequence divergence of 1.2%, differing by 8 amino acid residues (4 in prM and 4 in E) from those of PRVABC-59 (Figure 6B,C). Of these amino acid differences, the only common variations in the prM and E proteins between P6-740 and MR-766, as well as between P6-740 and PRVABC-59, are located at the following two sites (Figure 6C): (i) prM-1, the first-residue Ala-to-Val substitution in the prM protein of P6-740, located at the C-prM cleavage site processed by a cellular signal peptidase [125]. (ii) E-156, the 156th-residue Thr-to-Ile substitution in the E protein of P6-740, located at the N-linked glycosylation consensus site (^154^NDT) [126]. Therefore, our results suggest that either or both the Ala-to-Val substitution at the prM-1 site and the Thr-to-Ile substitution at the E-156 site are likely responsible for the unique virological properties of rJEV/ZIKV^P6-740^ in Vero cells and its more attenuated phenotypes in IFNAR^−/−^ mice when compared to rJEV/ZIKV^MR-766^ and rJEV/ZIKV^PRVABC-59^.

## 3. Discussion

In this study, we utilized our infectious cDNA technology based on the clinically proven, well-characterized live-attenuated JEV vaccine SA_14_-14-2 [117,118,121] to create three recombinant chimeric JEV/ZIKVs (rJEV/ZIKV^MR-766^, rJEV/ZIKV^P6-740^, and rJEV/ZIKV^PRVABC-59^) as candidate vaccine prototypes for ZIKV. In these chimeric viruses, the neutralizing antibody-inducing viral envelope genes, prM and E, of JEV SA_14_-14-2 were replaced with the corresponding genes from each of three genetically distinct ZIKV strains [23,82]: African MR-766, Asian P6-740, and American PRVABC-59. Comparison of these chimeric JEV/ZIKVs in Vero cells, a cell line suitable for vaccine production [124], and in IFNAR^−/−^ mice, an animal model for ZIKV infection [82], demonstrated the following: (i) The genetic makeup of the prM and E genes of ZIKV determines the distinctive in vitro virological properties of the three chimeric viruses. Notably, rJEV/ZIKV^P6-740^ exhibited the slowest viral growth, formed the smallest plaques, and displayed the highest prM-to-M protein expression ratio. (ii) The genetic composition of the prM and E genes of ZIKV confers differential attenuation phenotypes in vivo on the three chimeric viruses. Importantly, rJEV/ZIKV^P6-740^ caused a localized infection limited to the spleen, resulting in asymptomatic infection, whereas the other two chimeric viruses caused systemic infections involving multiple organs, leading to severe symptomatic infections and 100% mortality. (iii) A single-dose subcutaneous immunization with rJEV/ZIKV^P6-740^ induced ZIKV prM/E-specific neutralizing antibodies capable of providing complete protection against a challenge with each of the three genetically distinct ZIKV strains. (iv) The unique characteristics of rJEV/ZIKV^P6-740^, both in vitro and in vivo, as compared to rJEV/ZIKV^MR-766^ and rJEV/ZIKV^PRVABC-59^, are likely attributable to the Ala-to-Val substitution at the prM-1 site, the Thr-to-Ile substitution at the E-156 site, or both. Thus, our results indicate that the chimeric virus rJEV/ZIKV^P6-740^ is a highly promising vaccine prototype with the potential to be developed into a safe and effective vaccine that can induce protective immunity against ZIKV through neutralizing antibodies.

Since the ZIKV epidemic in 2015–2016, efforts to develop vaccines against the pathogen have rapidly accelerated [74,75,76]. To date, numerous vaccine candidates have been tested in pre-clinical and early clinical trials across various platforms, including killed-inactivated [111,127,128,129,130,131,132,133,134,135,136,137], live-attenuated [138,139,140,141,142], viral vector-based [108,143,144,145,146,147,148,149,150,151,152,153,154,155,156,157,158,159,160,161,162,163,164], DNA-based [105,111,131,165,166,167,168,169,170,171,172,173,174], mRNA-based [106,141,175,176,177,178], protein-based [169,179,180,181,182,183,184,185,186,187], and virus-like particle-based [188,189,190,191,192,193] vaccines. Among these, the killed-inactivated and live-attenuated vaccine candidates have been produced using whole virions of African MR-766 [132,188], Asian FSS13025 [138,140,141,142], or American GZ02 [139], MEX2-81 [130], or PRVABC-59 [111,127,128,129,131,133,134,135,136,137] strains. All the other vaccine candidates have been designed almost exclusively to express or deliver an immunogen of the prM and E proteins derived from a specific ZIKV strains(s) within the African lineage (MR-766 [189,191]), the Asian lineage (FSS13025 [143,152,154,155,157,186,187], H/PF/2013 [105,106,160,161,166,177,183,184,188], Micronesia 2007 [141,175], and 1_0080_PF [153]), or the American sublineage within the Asian lineage (BeH815744 [108,111,131,147], GZ01 [172], Martinique [145], Natal [162,163,164,177], PRVABC-59 [144,146,156,182,185,190,193], Rio-S1 [178], SPH2015 [148,149,167,174,176,179,189,191], SZ-WIV01 [192], Z1106033 [150,180], and ZKV17uri [181]), or those with a consensus sequence obtained from multiple ZIKV strains [158,165,170,171,173]. These studies demonstrate that neutralizing antibodies against the prM and E proteins of ZIKV provide protective immunity and are crucial indicators of protection against the pathogen. Therefore, the prM and E proteins are the primary target antigens for developing ZIKV vaccines.

We utilized the live-attenuated JEV vaccine SA_14_-14-2 as a genetic backbone to develop chimeric JEV/ZIKVs that express the neutralizing antibody-inducible prM and E proteins of ZIKV, serving as potential vaccine prototypes for ZIKV. This strategy offers several advantages, including strong and long-lasting immunity, a comprehensive immune response, no need for adjuvants, and a requirement for few doses [194,195]. Similar to our study, previous research has employed the same strategy to generate prM-E gene-replaced chimeric orthoflaviviruses as vaccine candidates for ZIKV: (i) the live-attenuated JEV vaccine SA_14_-14-2 as a backbone, replacing the prM and E genes with those of ZIKV FSS13025, an Asian strain isolated in Cambodia in 2010 [154], (ii) the live-attenuated YFV vaccine 17D as a backbone, replacing the prM and E genes with those of ZIKV H/PF/2013, an Asian strain isolated in French Polynesia in 2013 [160,161], (iii) the live-attenuated DENV vaccine PDK-53 as a backbone, replacing the prM and E genes with those of ZIKV PRVABC-59, an American strain isolated in Puerto Rico in 2015 [156], and (iv) the insect-specific orthoflavivirus Binjari virus as a backbone, replacing the prM and E genes with those of ZIKV Natal, an American strain isolated in Brazil in 2015 [162,163,164]. The results of these studies align with ours, indicating that the chimeric vaccine candidates are effective in inducing a high level of neutralizing antibody response and providing full protection against ZIKV challenge, potentially offering long-lasting immunity with a single-dose immunization. Hence, the chimeric vaccine development strategy is not only promising for developing vaccines against ZIKV but also applicable for developing new vaccines against other orthoflaviviruses [196]. However, there are conceivable safety issues regarding the use of prM and E genes from wild-type strains of ZIKV or other orthoflaviviruses to replace those of the fully attenuated JEV SA_14_-14-2, YFV 17D, DENV PDK-53, or the insect-specific Binjari virus. Indeed, there are 10 amino acid differences in the prM-E gene region (1 in prM and 9 in E) between JEV SA_14_-14-2 and its pre-attenuated wild-type strain SA_14_ [117]. Since the E protein is a key virulence factor for orthoflaviviruses [100,101], using the prM and E genes from a wild-type ZIKV strain to replace those of the fully attenuated JEV SA_14_-14-2 can raise potential safety concerns for chimeric vaccine development.

We hypothesized that the primary sequence of the prM and E genes of ZIKV would affect the biological attributes of chimeric JEV/ZIKVs. To test this hypothesis, we created three chimeric JEV/ZIKVs (rJEV/ZIKV^MR-766^, rJEV/ZIKV^P6-740^, and rJEV/ZIKV^PRVABC-59^) by replacing the prM and E genes of JEV SA_14_-14-2 with those from three genetically divergent ZIKV strains: African MR-766, Asian P6-740, and American PRVABC-59. Among the three chimeric viruses, rJEV/ZIKV^P6-740^ was distinctive, characterized by the slowest viral growth rate, smallest plaque formation, and highest prM-to-M protein expression ratio in Vero cells. In wild-type weanling C57BL/6J mice, all three chimeric viruses behaved like the vector rJEV. However, in age- and genetic background-matched IFNAR^−/−^ mice, rJEV/ZIKV^P6-740^ was more attenuated than the vector rJEV, while the other two chimeric viruses were attenuated similarly to the vector rJEV. This favorable attenuation phenotype of rJEV/ZIKV^P6-740^ as a live vaccine candidate was more evident when comparing viral loads in various organs after infection. In IFNAR^−/−^ mice, subcutaneous inoculation of rJEV/ZIKV^P6-740^ resulted in a low-level, localized infection in the spleen with no clinical signs. In contrast, the other two chimeric viruses and their vector caused severe systemic infections affecting multiple organs, leading to clear clinical signs and 100% mortality. Subsequent pairwise amino acid sequence comparison of the prM and E proteins among the three ZIKVs revealed two unique variations in P6-740 compared to MR-766 and PRVABC-59: (i) an Ala-to-Val substitution at amino acid position 1 in the prM protein of P6-740, located at the C-prM cleavage site processed by a cellular signal peptidase [125,197]. Since both Ala and Val are small, nonpolar, hydrophobic amino acids, the Ala-to-Val substitution is less likely to significantly alter the protein’s structure or function or the efficiency of the proteolysis at the C-prM cleavage site [198]. (ii) a Thr-to-Ile substitution at amino acid position 156 in the E protein of P6-740, located at the N-linked glycosylation consensus site (^154^NDT) [126]. This change has been shown to abolish the N-linked glycosylation of the E protein, potentially affecting the biological properties of rJEV/ZIKV^P6-740^ [199,200,201,202,203,204]. In this study, we found that the lack of glycosylation in the E protein of P6-740 was consistent with its faster mobility on a reducing gel compared to that of MR-766 and PRVABC-59. Further investigations are warranted to determine the genetic traits in the prM and E genes of P6-740 responsible for the unique biological attributes of rJEV/ZIKV^P6-740^ in Vero cells and IFNAR^−/−^ mice.

In summary, we have developed a recombinant chimeric virus, rJEV/ZIKV^P6-740^, as a vaccine prototype against ZIKV. This prototype is based on the genetic background of the live-attenuated JEV vaccine SA_14_-14-2 and expresses the functional prM and E proteins of the ZIKV strain P6-740, which induce neutralizing antibodies. We anticipate that this vaccine prototype can be further refined into a safe and effective vaccine for ZIKV, maximizing safety while maintaining immunogenicity and protective efficacy and capitalizing on a single-dose immunization to achieve strong and long-lasting immunity. Future studies should address several issues in more detail, including the risk of reversion to a more virulent form, long-term safety and efficacy in diverse populations (e.g., immunocompromised individuals, pregnant women, and children), and potential environmental impacts such as the risk of the vaccine virus being transmitted through mosquito vectors or other means. Also, challenges such as the complex regulatory approval process and the difficulties in scaling up production and ensuring equitable distribution, especially in low-resource settings, must be overcome. Addressing these limitations and challenges is essential for the successful development and deployment of the vaccine.

## 4. Materials and Methods

### 4.1. Cell Culture

WHO Vero RCB 10-87 (Vero) cells were cultured in α-minimal essential medium (MEM) supplemented with 10% fetal bovine serum (FBS) and 100 U/mL penicillin–streptomycin at 37 °C in a 5% CO_2_ atmosphere [82]. All cell culture media and reagents were obtained from Gibco, Carlsbad, CA, USA.

### 4.2. cDNA Cloning

Using our full-length infectious cDNA clone for JEV SA_14_-14-2 (designated pBac/JEV [121]), we cloned three full-length chimeric cDNA constructs: pBac/JEV/ZIKV^MR-766^, pBac/JEV/ZIKV^P6-740^, and pBac/JEV/ZIKV^PRVABC-59^. These constructs were produced by replacing the prM and E genes of JEV SA_14_-14-2 with the corresponding genes from each of the three ZIKV strains: MR-766, P6-740, and PRVABC-59 [82]. In each case, a ZIKV prM-E gene fragment, flanked by the entire JEV C gene and 5’NCR at the 5’ end and the 5’-terminal portion of the JEV NS1 gene at the 3’ end, was synthesized using an overlapping extension PCR method. Initially, two DNA fragments were amplified by PCR using the following cDNA templates and primer pairs: (i) pBac/JEV cDNA with Vector-1F + MR-2R primers and pBac/ZIKV MR-766 cDNA with MR-3F + MR-4R primers for the construction of pBac/JEV/ZIKV^MR-766^, (ii) pBac/JEV cDNA with Vector-1F + P6-2R primers and pBac/ZIKV P6-740 cDNA with P6-3F + P6-4R primers for the construction of pBac/JEV/ZIKV^P6-740^, and (iii) pBac/JEV cDNA with Vector-1F + PRVABC-2R primers and pBac/ZIKV PRVABC-59 cDNA with PRVABC-3F + PRVABC-4R primers for the construction of pBac/JEV/ZIKV^PRVABC-59^. In each case, the two amplicons were fused by a second round of PCR using the following primers: Vector-1F + MR-4R for pBac/JEV/ZIKV^MR-766^, Vector-1F + P6-4R for pBac/JEV/ZIKV^P6-740^, and Vector-1F + PRVABC-4R for pBac/JEV/ZIKV^PRVABC-59^. The resulting fused amplicons were then used to replace the corresponding region in pBac/JEV using *Pac*I and *Ssp*I, creating pBac/JEV/ZIKV^MR-766^, pBac/JEV/ZIKV^P6-740^, and pBac/JEV/ZIKV^PRVABC-59^. All cDNA cloning work was performed using standard molecular cloning techniques. The oligonucleotide primers are summarized in Table 1. For our cDNA cloning experiments, we obtained oligonucleotides from Integrated DNA Technologies (San Diego, CA, USA), Advantage HD polymerase from Clontech (Mountain View, CA, USA), and all restriction enzymes from New England Biolabs (Ipswich, MA, USA).

### 4.3. Infectious Center Assay

For in vitro run-off transcription, a full-length cDNA construct of JEV, ZIKV, or chimeric JEV/ZIKV was linearized using *Xba*I digestion followed by mung bean nuclease treatment. The linearized plasmid was then purified via phenol–chloroform extraction and ethanol precipitation. This purified plasmid served as the DNA template for in vitro run-off transcription, as previously reported [121,205]. The transcription reaction was carried out with SP6 RNA polymerase in the presence of the RNA cap analog m^7^G(5’)ppp(5’)A at 37 °C for 1 h, according to the manufacturer’s instructions. All enzymes and reagents for the in vitro run-off transcription were purchased from New England Biolabs. After transcription, the transcript quality was evaluated using 0.6% agarose gel electrophoresis and a NanoDrop 2000 spectrophotometer (Thermo Scientific, Waltham, MA, USA).

An infectious center assay, which measures the specific infectivity of in vitro synthesized run-off RNA transcripts from a linearized full-length cDNA, involves introducing the RNA into Vero cells via electroporation using an ECM 830 electroporator (BTX, San Diego, CA, USA), as previously described [82,205]. In brief, Vero cells were harvested by trypsin digestion, washed three times with cold phosphate-buffered saline (PBS), and resuspended in the same cold PBS at a density of 2 × 10^7^ cells/mL. A 400 µL aliquot of the resuspended cells was electroporated with 2 µg of RNA using the following settings: 980 volts, 99 µs pulse length, and 3 pulses. After a 10 min incubation at room temperature, 10-fold serial dilutions of the electroporated cells were placed onto monolayers of non-electroporated cells (5 × 10^5^ cells/well) seeded in a six-well plate. Following a 4 to 6 h incubation, the cells were overlaid with 3 mL of α-MEM containing 10% FBS and 0.5% SeaKem LE agarose (FMC BioProducts, Rockland, ME). After 5 days of incubation, the cell monolayers were fixed with 7% formaldehyde, and infectious centers of plaques were immunostained (see below for a detailed description).

### 4.4. Recovery of Recombinant Viruses from Functional cDNAs

Vero cells were electroporated with 10 µg of run-off RNA transcripts synthesized in vitro from a full-length cDNA, as described above. The transfected cells were seeded into a 150 mm culture dish at a density of 4 × 10^6^ cells/dish and cultured for 1–2 days in complete culture medium until ~75% of the cells died. The culture supernatant containing recombinant viruses was then collected. Cell debris in the collected supernatant was removed by centrifugation at 4000 rpm for 10 min at 4 °C. The cleared supernatant was stored at −80 °C until use.

### 4.5. Immunoblot Analysis

Viral proteins expressed in Vero cells were detected by immunoblotting, as previously described [206]. In brief, cells were lysed using a buffer containing 80 mM Tris–HCl (pH 6.8), 2% sodium dodecyl sulfate (SDS), 10% glycerol, 100 mM dithiothreitol, and 0.2% bromophenol blue. Equal volumes of the lysates were boiled for 5 min, separated on an SDS–polyacrylamide gel using a Mini Protean Vertical Electrophoresis System (Bio-Rad, Hercules, CA, USA), and transferred to a polyvinylidene difluoride membrane with a Trans-Blot SD Semi-Dry Transfer Cell (Bio-Rad). The membrane was stained with one of the following polyclonal rabbit antisera at a 1:500 to 1:1000 dilution: α-_J_C (specific to JEV C), α-_J_M (JEV M), α-_J_E (JEV E), α-_J_NS1 (JEV NS1), α-_Z_M (ZIKV M), and α-_Z_E (ZIKV E) [82,206]. Antibody-reactive proteins were detected by incubation with alkaline phosphatase (AP)-conjugated goat α-rabbit IgG (Jackson ImmunoResearch, West Grove, PA, USA) at a 1:1000 dilution, followed by reaction with a solution of 5-bromo-4-chloro-3-indolyl phosphate and nitroblue tetrazolium (Sigma-Aldrich, St. Louis, MO, USA) as substrates for the AP.

### 4.6. Viral Growth Analysis

Vero cells, seeded in a 35 mm culture dish at a density of 3 × 10^5^ cells/dish for 15 h, were infected with JEV, ZIKV, or chimeric JEV/ZIKV at an MOI of 1 in a final volume of 1.5 mL for 1 h with frequent agitation. The cell monolayers were washed once with α-MEM and incubated in 5 mL of compete culture medium. At 6, 12, 18, 24, 36, 48, 60, and 72 hpi, a portion of the culture medium was harvested in duplicate for virus titration. For the titration, Vero cells, seeded in a 6-well plate at a density of 3 × 10^5^ cells/well for 15 h, were infected with serial 10-fold dilutions of the collected culture medium in a final volume of 1 mL for 1 h. The cell monolayers were then overlaid with 3 mL of α-MEM containing 10% FBS and 0.5% SeaKem LE agarose (FMC BioProducts). At 5 days post-infection, viral plaques were identified by immunostaining (see below for a detailed description). Viral titers are presented as PFUs per mL.

### 4.7. Plaque Immunostaining (Immunoplaque Assay)

Viral plaques formed on Vero cells were immunostained as previously described [82,121]. In brief, the cell monolayer was fixed with 7% formaldehyde, permeabilized with 0.25% Triton X-100, and then stained with a polyclonal rabbit α-JEV NS3 [206] or α-ZIKV NS1 [82] antiserum at a 1:1000 dilution. Antibody-reactive plaques were detected by incubation with horseradish peroxidase (HRP)-conjugated goat α-rabbit IgG (Jackson ImmunoResearch) at a 1:1000 dilution, followed by reaction with a solution of 3,3’-diaminobenzidine (Vector Laboratories, Burlingame, CA, USA) as a substrate for the HRP.

### 4.8. Mouse Studies

This study utilized two mouse strains: (i) C57BL/6J, purchased from Charles River Laboratories, Wilmington, MA, and (ii) IFNAR^−/−^, produced on the C57BL/6J genetic background and maintained in our animal facility [82]. Groups of 4-week-old male and female mice were inoculated subcutaneously or intramuscularly with 1.5 × 10^5^ PFUs/mouse of JEV, ZIKV, or chimeric JEV/ZIKV in a total volume of 50 µL in α-MEM. For control, a group of mice was mock-inoculated with the same volume of Vero cell culture supernatant, either subcutaneously or intramuscularly. After inoculation, the mice were monitored daily for clinical signs, body weight changes, and survival rates over 22 days. Survival rates were analyzed using the Kaplan–Meier method, and body weight changes are expressed as percentages, with the pre-infection weight set to 100%.

### 4.9. Ethics Statement

All mouse studies were conducted in Animal Biosafety Level (ABSL)-2 and ABSL-3+ facilities at the Bioinnovations Campus of Utah State University (USU). These experiments strictly adhered to the National Institutes of Health guidelines for the care and use of laboratory animals. The protocol was approved by the USU Institutional Animal Care and Use Committee (protocol number 10109; approval date 10 October 2018). Mice were handled minimally and euthanized when moribund to minimize discomfort, distress, pain, and injury.

### 4.10. Plaque Reduction Neutralization Test (PRNT)

The PRNT was used to titrate neutralizing antibodies in pre- and post-immune serum samples from the mice. Serum samples were heat-inactivated at 56 °C for 30 min and then serially diluted two-fold from 1:32 to 1:2048. Fifty microliters of the diluted serum were mixed with an equal volume of 100 PFUs of rZIKV MR-766, rZIKV P6-740, rZIKV PRVABC-59, or rJEV SA_14_ and incubated for 1 h at 37 °C. In parallel, the same amount of each virus was incubated without serum as a control. The mixture was then placed onto Vero cells seeded in a 6-well plate (3 × 10^5^ cells/well) and incubated for 1 h at 37 °C with frequent agitation. After incubation, the cells were overlaid with 3 mL of α-MEM containing 10% FBS and 0.5% SeaKem LE agarose (FMC BioProducts) and incubated for 5 days at 37 °C in a 5% CO_2_ incubator. Viral plaques were visualized by immunostaining (see above for a detailed description). The neutralizing antibody titer was estimated as the reciprocal of the highest dilution that reduced plaque counts by at least 50%, compared to the virus-only control, and presented as log_10_ PRNT_50_ values.

### 4.11. Sequence Analysis

The complete genome sequences for the four analyzed viruses are available in GenBank: JEV SA_14_-14-2 (JN604986), ZIKV MR-766 (KX377335), ZIKV P6-740 (KX377336), and ZIKV PRVABC-59 (KX377337). Signal peptide cleavage sites at the C-prM and E-NS1 junctions in the viral polyprotein of the three ZIKV strains were predicted using the SignalP program [122], accessible at https://services.healthtech.dtu.dk/services/SignalP-4.1/, accessed on 25 June 2018. The amino acid sequences at these junctions were compared with those of JEV SA_14_-14-2 using the BLAST blastp program, available at https://blast.ncbi.nlm.nih.gov/Blast.cgi, accessed on 25 June 2018. A multiple amino acid sequence alignment of the prM-E gene segment of the three ZIKV strains was generated using the Clustal W method [123].

### 4.12. Statistical Analysis

Mouse body weight changes were represented as the ratio of weights at various times *t* to their initial weights at time *t* = 0. These weight ratios were analyzed using a mixed-effects model, with group, log(time), and their interaction as fixed effects and mouse identity as a random effect. A radial smoother was included to account for the temporal correlation in the repeatedly measured weights. Instead of pairwise comparisons, contrasts of clinical interest were tested using ESTIMATE statements. Weight analyses were performed using PROC GLIMMIX from SAS/STAT 15.3 (SAS Institute Inc., Cary, NC, USA). The analysis of survival rate and the estimation of survival time were performed using PROC LIFETEST from the same software. Statistical significance was set at the 0.05 level.

## Figures and Tables

**Figure 1 ijms-26-00195-f001:**
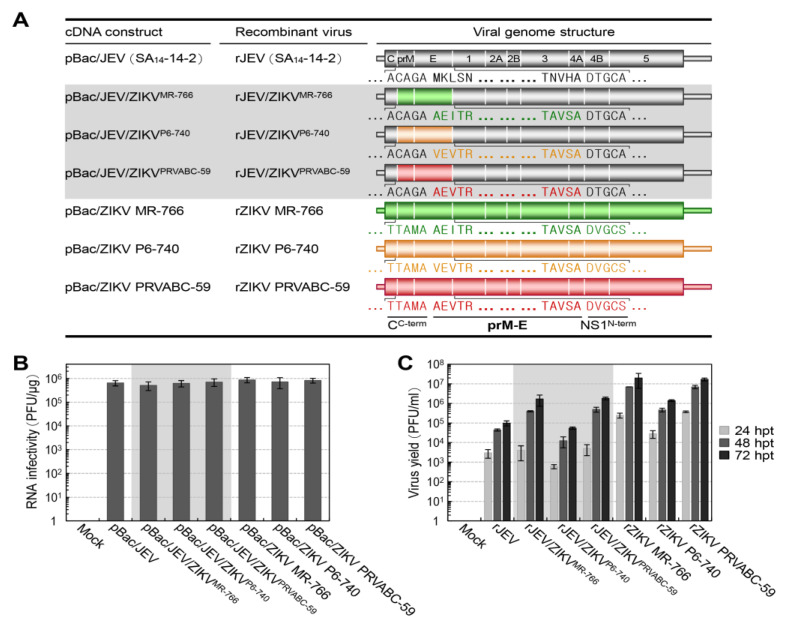
Creation of prM-E gene-replaced chimeric JEV/ZIKVs. (**A**) Schematic illustration of the genomes of the parental JEV, three original ZIKVs, and three chimeric JEV/ZIKVs. The molecularly cloned recombinant viruses were rescued from each full-length cDNA construct as indicated. Viral genome structures are depicted, highlighting the amino acid sequences at the C-prM and E-NS1 cleavage sites. (**B**,**C**) Recovery of recombinant viruses from the full-length JEV, ZIKV, and JEV/ZIKV cDNA constructs. Capped run-off RNA transcripts synthesized in vitro from each full-length cDNA construct were transfected into Vero cells. Mock-transfected cells served as a negative control. The level of RNA infectivity was determined as PFU per µg by infectious center assays (**B**). The level of virus accumulation in culture supernatants collected at 24, 48, and 72 h post-transfection (hpt) was determined as PFU per mL by immunoplaque assays on Vero cells (**C**).

**Figure 2 ijms-26-00195-f002:**
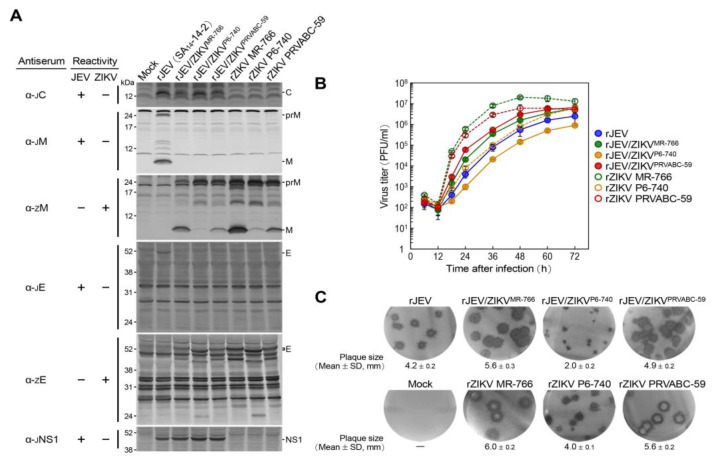
In vitro virological properties of prM-E gene-replaced chimeric JEV/ZIKVs in Vero cells. Vero cells were either mock-infected or infected with each virus at an MOI of 1. (**A**) Viral protein expression: At 24 h post-infection, total cell lysates were analyzed by immunoblotting with the respective rabbit antisera. (**B**) Viral growth kinetics: Culture supernatants were collected at the indicated time points, and virus accumulation levels were determined by immunoplaque assays on Vero cells. (**C**) Viral plaque morphology: At 5 days post-infection, cell monolayers were immunostained with α-JEV NS3 or α-ZIKV NS1 antiserum, as appropriate, to visualize the plaques. Plaque sizes were measured by determining the diameter of 10 randomly selected plaques and presented as mean values with standard deviations (mean ± SD).

**Figure 3 ijms-26-00195-f003:**
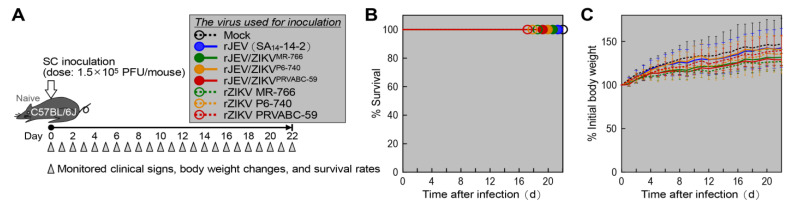
In vivo non-pathogenic nature of prM-E gene-replaced chimeric JEV/ZIKVs in C57BL/6J mice. (**A**) Experimental scheme: Groups of 4-week-old mice (*n* = 6, 3 males and 3 females) were either mock-infected or infected subcutaneously with 1.5 × 10^5^ PFUs/mouse of each virus as indicated. The mice were monitored daily for clinical signs, body weight changes, and survival rates. (**B**) Survival curves: Survival rates were analyzed using the Kaplan–Meier method. (**C**) Body weight changes: Body weight changes are shown as percentages, with the weight on the day before infection set as the baseline (100%). Data are presented as means ± standard deviations.

**Figure 4 ijms-26-00195-f004:**
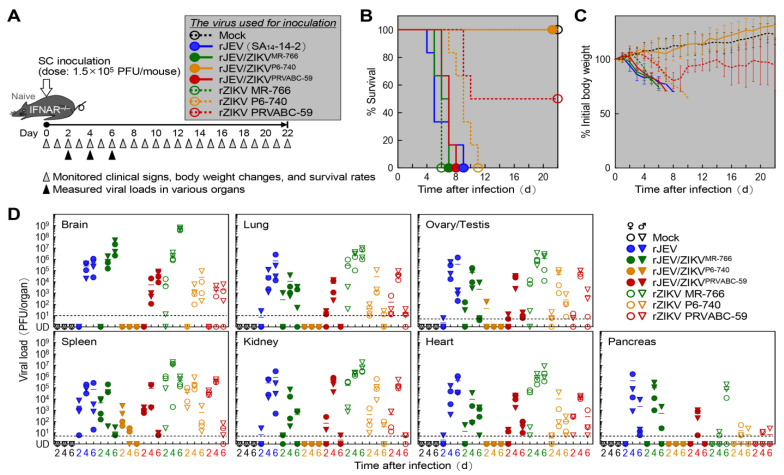
In vivo attenuation phenotypes of prM-E gene-replaced chimeric JEV/ZIKVs in IFNAR^−/−^ mice. (**A**) Experimental scheme: Groups of 4-week-old mice (*n* = 6, 3 males and 3 females) were either mock-infected or infected subcutaneously with 1.5 × 10^5^ PFUs/mouse of each virus as indicated. The mice were monitored daily for clinical signs, body weight changes, and survival rates. (**B**) Survival curves: Survival rates were analyzed using the Kaplan–Meier method. (**C**) Body weight changes: Body weight changes are shown as percentages, with the weight on the day before infection set as the baseline (100%). Data are presented as means ± standard deviations. (**D**) Viral loads in organs: As described in (**A**), groups of 4-week-old mice (*n* = 12, 6 males and 6 females) were either mock-infected or infected subcutaneously with 1.5 × 10^5^ PFUs/mouse of each virus. At 2, 4, and 6 days post-infection, organs were harvested from four infected mice (2 males and 2 females) per group. Viral loads in each organ were determined by immunoplaque assays on Vero cells. The dotted line indicates the limit of detection.

**Figure 5 ijms-26-00195-f005:**
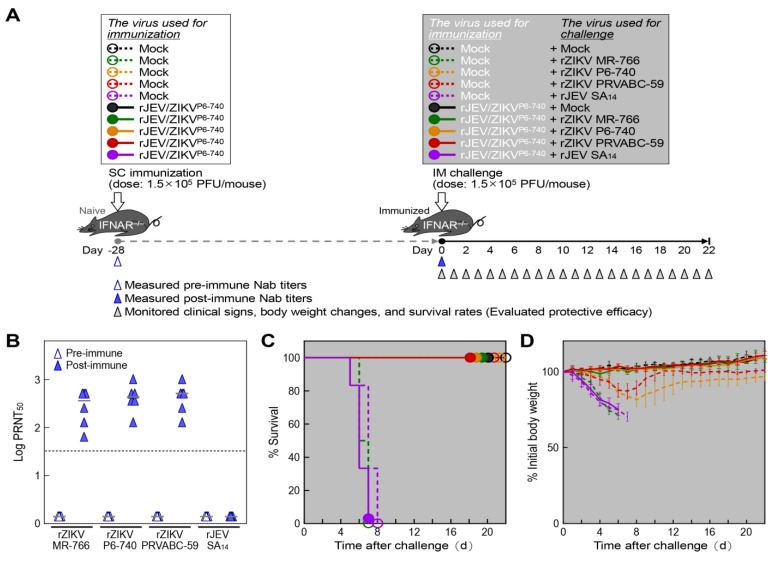
Protective efficacy of the chimeric virus rJEV/ZIKV^P6-740^ against ZIKV in IFNAR^−/−^ mice. (**A**) Experimental scheme: Groups of 4-week-old mice (*n* = 6, 3 males and 3 females) were either mock-immunized or immunized subcutaneously with 1.5 × 10^5^ PFUs/mouse of rJEV/ZIKV^P6-740^. On day 28 post-immunization, the mice were either mock-challenged or challenged intramuscularly with 1.5 × 10^5^ PFUs/mouse of rZIKV MR-766, P6-740, PRVABC-59, or rJEV SA_14_. The mice were monitored daily for clinical signs, body weight changes, and survival rates. Blood samples were drawn before immunization (pre-immune) and after immunization but before the challenge (post-immune). (**B**) Neutralizing antibody (Nab) titers: Nab titers were determined as log_10_ PRNT_50_ values using pre- and post-immune antisera on Vero cells with rZIKV MR-766, P6-740, PRVABC-59, or rJEV SA_14_. The dotted line indicates the limit of detection. (**C**) Survival curves: Survival rates were analyzed using the Kaplan–Meier method. (**D**) Body weight changes: Body weight changes are shown as percentages, with the weight on the day before challenge set as the baseline (100%). Data are plotted with means ± standard deviations.

**Figure 6 ijms-26-00195-f006:**
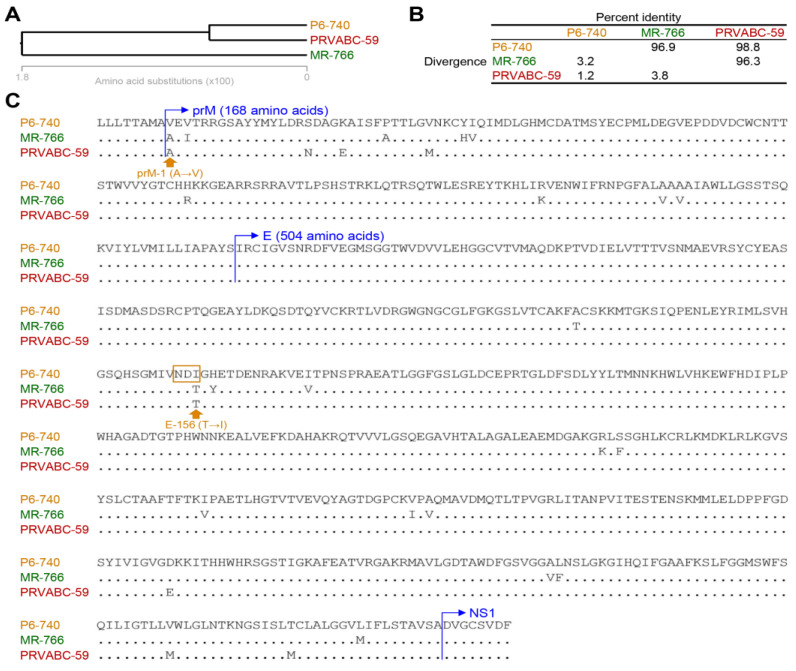
Amino acid sequence comparison of the prM and E proteins of three ZIKV strains. A multiple sequence alignment was performed using the amino acid sequences of the prM and E proteins from three ZIKV strains (MR-766, P6-740, and PRVABC-59). The alignment outputs include (**A**) the unrooted phylogenetic tree, depicting the evolutionary relationships among the three ZIKV strains; (**B**) sequence identity and divergence (%), showing the identity and divergence of the amino acids in the prM and E proteins; and (**C**) amino acid variation, indicating the specific amino acid differences among the three ZIKV strains. The N-termini of the prM and E proteins are shown in blue. Two unique amino acid differences (prM-1 and E-156) present in P6-740, as compared to MR-766 and PRVABC-59, are indicated in orange. A box highlights the N-linked glycosylation site in the E protein.

**Table 1 ijms-26-00195-t001:** Sequences of the oligonucleotides used in this study.

**Oligonucleotide**	**Sequence** (**5’→3’**)	**Direction**
Vector-1F	GGCATACCCCGCGTATTCCCACTA	Forward
MR-2R	TCTAGTGATCTCTGCGGCTCCTGCACAAGCTATGACA	Reverse
MR-3F	GCTTGTGCAGGAGCCGCAGAGATCACTAGACGCGGGA	Forward
MR-4R	GCAAATATTTATACCTATCCACCCAGGCTTCCACGTCGTTGTGCACGAAGATGCCACTTCCACATCTCATCTCTTTTCTTGTGATGTCAATGGCACATCCAGTGTCAGCAGAAACAGCCGTGGAGAGGAAGATCAT	Reverse
P6-2R	TCTGGTGACCTCCACGGCTCCTGCACAAGCTATGACA	Reverse
P6-3F	GCTTGTGCAGGAGCCGTGGAGGTCACCAGACGTGGGA	Forward
P6-4R	GCAAATATTTATACCTATCCACCCAGGCTTCCACGTCGTTGTGCACGAAGATGCCACTTCCACATCTCATCTCTTTTCTTGTGATGTCAATGGCACATCCAGTGTCAGCAGAGACGGCTGTAGATAGGAAGATCAA	Reverse
PRVABC-2R	TCTAGTGACCTCCGCGGCTCCTGCACAAGCTATGACA	Reverse
PRVABC-3F	GCTTGTGCAGGAGCCGCGGAGGTCACTAGACGTGGGA	Forward
PRVABC-4R	GCAAATATTTATACCTATCCACCCAGGCTTCCACGTCGTTGTGCACGAAGATGCCACTTCCACATCTCATCTCTTTTCTTGTGATGTCAATGGCACATCCAGTGTCAGCAGAGACGGCTGTGGATAAGAAGATCAA	Reverse

## Data Availability

The original contributions presented in this study are included in the article. Further inquiries can be directed to the corresponding author.

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
