# Peer review of "Comparison of Three Chimeric Zika Vaccine Prototypes Developed on the Genetic Background of the Clinically Proven Live-Attenuated Japanese Encephalitis Vaccine SA14-14-2"

_ijms, 2024, doi:10.3390/ijms26010195_

Round 1

Reviewer 1 Report

Comments and Suggestions for Authors

This manuscript [“Comparison of Three Chimeric Zika Vaccine Prototypes Developed on the Genetic Background of the Clinically Proven Live-Attenuated Japanese Encephalitis Vaccine SA14-14-2”] by Song et al. investigated three recombinant chimeric viruses developed as candidate vaccine prototypes for Zika virus (ZIKV) infection. Their best candidate among the recombinant chimeric viruses was the rJEV/ZIKVP6-740, which showed great potential as a vaccine prototype against ZIKV. The study in overall is interesting, the manuscript is well written, and the results are quite promising. However, I have a few minor comments/suggestions to the authors before I can recommend it for the publication in the International Journal of Molecular Sciences.

My suggestions/comments are enlisted below.

1)      Fig. 1 (C): It would be nice to have error bars for all the datasets. It is important for the comparison of different samples and to firmly establish the claims made by the authors.

2)      Fig. 2 (B): Again, it is important to include the error bars for proper comparison of different samples.

3)      Plaque sizes: Please include error bars for these values from multiple measurements (with n value).

Reviewer 2 Report

Comments and Suggestions for Authors

Thank you for the opportunity to review this article by Song et al, describing the development of 3 prototype chimeric JEV/ZIKV vaccines against Zika virus infection. The authors used well established methods to develop the vaccine and test its properties in vitro and in vivo in mice. The methodology is sound and well-described. The manuscript is well-written and the introduction sufficiently illustrates the scientific background of the study. The results are presented in a coherent stepwise manner, and the discussion section appropriately elaborates on the strengths of this approach. I have only one comment:

1. What are the limitations of this study and what difficulties do the authors anticipate for their prototype vaccine entering further steps of evaluation. 

Reviewer 3 Report

Comments and Suggestions for Authors

The manuscript presents the development and evaluation of three recombinant chimeric viruses as candidate vaccines for Zika virus (ZIKV). These chimeric viruses were constructed by replacing the prM and E genes of the Japanese encephalitis virus (JEV) vaccine strain SA14-14-2 with those from three genetically distinct ZIKV strains. The study primarily investigates the viral growth characteristics, immunogenicity, and protective efficacy of these chimeric viruses, with a focus on the prototype virus, rJEV/ZIKVP6-740. The results indicate that rJEV/ZIKVP6-740 exhibits favorable properties for use as a vaccine candidate, showing slow viral growth, reduced plaque size, and a unique protein expression profile. Furthermore, immunization with rJEV/ZIKVP6-740 provided complete protection against a lethal ZIKV challenge in an animal model, which is a significant finding for the development of a safe and effective ZIKV vaccine.

Specific Comments for Consideration in the Manuscript:

  1. Statistical Analysis: Several datasets in the manuscript lack statistical analysis or do not specify the statistical methods used. It is imperative that appropriate statistical tests be applied to all relevant data. Additionally, a dedicated section outlining the statistical methodologies should be incorporated into the Materials and Methods section.

  2. Immune Response Characterization: The manuscript would benefit from a more comprehensive evaluation of the vaccine's immunological profile, including an assessment of T cell responses, cytokine secretion, and long-term immunity following the challenge. These data would provide deeper insights into the vaccine’s mechanisms of immune protection.

  3. Wild-Type Mouse Testing: It is recommended that the rJEV/ZIKVP6-740 vaccine candidate be tested in wild-type mice. While rJEV/ZIKVP6-740 and rJEV display similar behavior in Vero cells, there is a notable discrepancy in their performance in IFNAR–/– mice. This discrepancy warrants further investigation into the immune response and viral dynamics of rJEV/ZIKVP6-740 in a wild-type mouse model.

  4. Viral Load Assessment: Has the viral load been measured in different tissues following challenge to assess the virus distribution and clearance, similar to the attenuation phenotype analysis? If not, it is recommended to include this analysis to evaluate the virus's ability to replicate and clear from various tissues, which is crucial for understanding the vaccine's efficacy in controlling infection.

  5. Effectiveness Against Parental Zika Virus: The observation that rJEV/ZIKVP6-740-immunized mice show no protective efficacy against the parental wild-type Zika virus (Figure 5C) raises concerns. Does this suggest that the vaccine prototype may not provide protection against wild-type Zika virus infections, and if so, what implications does this have for its broader effectiveness and potential use in humans?

Round 2

Reviewer 3 Report

Comments and Suggestions for Authors

The authors have addressed all my queries.